# Melanogenesis Inhibitors from the Rhizoma of *Ligusticum Sinense* in B16-F10 Melanoma Cells In Vitro and Zebrafish In Vivo

**DOI:** 10.3390/ijms19123994

**Published:** 2018-12-11

**Authors:** Min-Chi Cheng, Tzong-Huei Lee, Yi-Tzu Chu, Li-Ling Syu, Su-Jung Hsu, Chia-Hsiung Cheng, Jender Wu, Ching-Kuo Lee

**Affiliations:** 1School of Pharmacy, Taipei Medical University, Taipei 11031, Taiwan; d301100008@tmu.edu.tw; 2Institute of Fisheries Science, National Taiwan University, Taipei 10617, Taiwan; thlee1@ntu.edu.tw; 3Graduate Institute of Pharmacognosy, Taipei Medical University, Taipei 11031, Taiwan; m303098006@tmu.edu.tw (Y.-T.C.); goat1201@hotmail.com (L.-L.S.); 4Department of Food Science, National Chiayi University, Chiayi 60004, Taiwan; r1101815@yahoo.com.tw; 5Department of Biochemistry and Molecular Cell Biology, School of Medicine, College of Medicine, Taipei Medical University, Taipei 11031, Taiwan; chcheng@tmu.edu.tw; 6Ph.D. Program in Biotechnology Research and Development, Taipei Medical University, Taipei 11031, Taiwan

**Keywords:** *Ligusticum sinense*, rhizoma, antimelanogenesis, B16-F10 melanoma cell, zebrafish, pigmentation

## Abstract

The rhizoma of *Ligusticum sinense*, a Chinese medicinal plant, has long been used as a cosmetic for the whitening and hydrating of the skin in ancient China. In order to investigate the antimelanogenic components of the rhizoma of *L. sinense*, we performed an antimelanogenesis assay-guided purification using semi-preparative HPLC accompanied with spectroscopic analysis to determine the active components. Based on the bioassay-guided method, 24 compounds were isolated and identified from the ethyl acetate layer of methanolic extracts of *L. sinense*, and among these, 5-[3-(4-hydroxy-3-methoxyphenyl)allyl]ferulic acid (**1**) and *cis*-4-pentylcyclohex-3-ene-1,2-diol (**2**) were new compounds. All the pure isolates were subjected to antimelanogenesis assay using murine melanoma B16-F10 cells. Compound **1** and (3*S*,3a*R*)-neocnidilide (**8**) exhibited antimelanogenesis activities with IC_50_ values of 78.9 and 31.1 μM, respectively, without obvious cytotoxicity. Further investigation showed that compound **8** demonstrated significant anti-pigmentation activity on zebrafish embryos (10‒20 μM) compared to arbutin (20 μM), and without any cytotoxicity against normal human epidermal keratinocytes. These findings suggest that (3*S*,3*aR*)-neocnidilide (**8**) is a potent antimelanogenic and non-cytotoxic natural compound and may be developed potentially as a skin-whitening agent for cosmetic uses.

## 1. Introduction

Melanin is a brown black pigment that is principally responsible for the color of skin, hair, and eyes [1]. Melanin is a biopolymer and includes two major classes of pigments in human skin, brownish black eumelanin and reddish yellow pheomelanin [1]. Melanin biosynthesis is a complex multistep process called melanogenesis, which is a physiological response of human skin to prevent deleterious effects of ultraviolet (UV) radiation and environmental pollutants. However, over-melanogenesis can lead to the darkening of the skin, and abnormal hyperpigmentation causes various dermatological problems, such as freckles, melasma, senile lentigines, and even skin cancer [2].

Melanin is produced in membrane-bound organelles referred to as melanosomes, which are present in specialized cells called melanocytes. Melanin synthesis starts from the hydroxylation of L-tyrosine to L-dihydroxyphenylalanin (DOPA) and is followed by oxidation to dopaquinone. The two reactions are catalyzed by a rate-limiting enzyme tyrosinase. In the absence of thiol substances, dopaquinone cyclizes to leukodopachrome, followed by a series of oxidoreduction reactions which involve tyrosinase-related protein-2 (Tyrp-2) to produce the intermediate dopachrome and 5,6-dihydroxyindole-2-carboxylic acid (DHICA). DHICA undergoes subsequent oxidation catalyzed by Tyrp-1 and polymerization to form eumelanin. Dopaquinone also conjugates with cysteine and glutathione to yield cysteinyldopa and glutathionyldopa, which are progressively transformed into pheomelanin [1].

The cells surrounding melanocytes such as keratinocytes and fibroblasts are involved in the regulation of melanogenesis [3]. Continuous exposure to UV irradiation induces DNA damages in keratinocytes and leads to p53-mediated up-regulation of proopiomelanocortin (POMC). POMC undergoes posttranslational cleavage to produce the melanocyte stimulating hormone (α-MSH) and β-endorphin. In turn, α-MSH binds to the melanocortin 1 receptor (MC1R) on adjacent melanocytes, resulting in the upregulation of cAMP. Elevated cAMP stimulates expression of microphthalmia-associated transcription factor (MITF). MITF then regulates the transcription of pigmentation enzymes, including tyrosinase, Tyrp-1 and Tyrp-2. The UV-triggered pathway eventually leads to melanin synthesis and transfer of melanosomes to keratinocytes [4,5,6]. In addition to extrinsic stimuli, melanin production is also influenced by intrinsic factors, such as hormone change, genetic disorders, inflammation, and age [3].

In East Asia, most women expect to avoid uneven skin pigmentation and pursue skin lightening. For instance, arbutin, kojic acid, azelaic acid, ascorbic acid, and white mulberry and licorice root extract have been used as whitening ingredients in cosmetic preparations [7]. The exploitation of effective and preventive skin whitening agents from natural sources is of great interest in the cosmetic field, primarily due to relative nontoxicity and fewer side effects [7,8]. The rhizoma of *Ligusticum sinense* Oliv. (Umbelliferae) have long been used as traditional Chinese medicine for 2000 years and till today its roots are a highly recommended herbal tea [9]. *L. sinense*, namely “Gaoben” in Chinese, also known as Chinese lovage, was used for expelling wind-cold, relieving pain and rheumatic arthralgia, and alleviating anemofrigid headache. The main external use of *L. sinense* is for skin whitening and hydrating [10]. To date, the reported chemical constituents present in *L. sinense* include terpenoids, phthalide analogues, and phenylpropanoid glycosides [11,12,13,14,15,16]. Such constituents have been reported to exert numerous pharmacological effects. For example, the essential oils from the roots and rhizomes of *L. sinense* were reported to possess analgesic, sedative, and antimicrobial effects [15,17]. Ligustilide, a main phthalide widely found in Umbelliferae plant, demonstrated dilatory effect on myometrium, and reduced inflammatory and neurogenic pain [18]. Cnidilide, another phthalide abundant in Umbelliferae plant, was proven to possess antispasmodic and inflammatory effects [19,20]. Previous studies have indicated that the extract of *L. sinense* inhibited melanogenesis on B16-F10 murine melanoma cells [21]. However, the active components for melanogenesis inhibitory activity were still unreported.

In our preliminary biological screening, it was found that the methanolic extracts of *L. sinense* exhibited antimelanogenesis activity in B16-F10 cells with an IC_50_ value of 50 μg/mL [22], and the antimealnogenesis principles are still undisclosed thus far. We thus set out to investigate the active principle of the rhizoma of *L. sinense* by a bioassay-directed method, and that has led to the isolation and identification of two new compounds **1** and **2** along with 22 known compounds **3**–**24**. This article also aimed to investigate the effects of compounds **1** and **8** on B16-F10 melanoma cells in vitro and zebrafish in vivo, assess the safety by normal human epidermal keratinocyte MTT assay, and quantify **1** and **8** in the rhizoma of *L. sinense*.

## 2. Results and Discussions

### 2.1. Isolation and Structural Elucidation 

In an attempt to isolate and identify the melanogenesis inhibitors efficiently from the active fractions, we employed a bioassay-guided fractionation strategy. A methanolic extract of the rhizoma of *L. sinense* was partitioned to give ethyl acetate, *n*-butanol and water soluble layers. The obtained three layers were then tested for antimelanogenesis activity in murine melanoma B16-F10 cells. The mouse B16 melanoma cell is a sensitive, reliable, and feasible platform for screening large number of small molecular melanogenesis regulators [23,24,25]. At the concentrations of 25–100 μg/mL, the ethyl acetate-soluble layer demonstrated the most potent inhibitory activity, while slight inhibition was observed for either the *n*-butanol or water-soluble layers (Figure 1A) as evidenced by melanin contents in lyzed B16-F10 melanoma cells (Figure 1B). Open column separation of ethyl acetate layer over silica gel followed by HPLC purification afforded two previously unreported chemical entities **1** and **2** (Figure 2A) together with 22 known compounds. The known compounds were characterized to be eugenol (**3**) [26], 2-hydroxy-4-methylacetophenone (**4**) [27], 3*S**,3a*R**,7a*S**-3-butylhexahydrophthalide (**5**) [28], carvacrol (**6**) [29], squalene (**7**) [30], (3*S*,3a*R*)-neocnidilide (**8**) [31], coniferyl alcohol 9-methylester (**9**) [32], bergapten (**10**) [33], methoxsalen (**11**) [34], methyl vanillate (**12**) [35], 2,5-dihydroxy-4-methylacetophenone (**13**) [36], 2-methoxy-4-nitrophenol (**14**) [37], 2,6-dimethoxyphenol (**15**) [38], falcarindiol (**16**) [39], (9*Z*)-heptadecene-4,6-diyne-1,8-diol (**17**) [40], 3-*O*-(*p*-coumaroyl)ursolic acid (**18**) [41], pregnenolone (**19**) [42], (9*Z*,11*E*,13*R*)-13-hydroxyoctadeca-9,11-dienoic acid (**20**) [43], ferulic acid (**21**) [44], coniferyl ferulate (**22**) [45], *p*-hydroxyphenethyl ferulate (**23**) [46], and vanillic acid (**24**) [47].

Compound **1**, obtained as colorless oil, had a formula of C_20_H_20_O_6_ as deduced from ^13^C NMR and positive-ion HRESI-MS, which showed a fragment ion at positive HRESI-MS *m/z* 357.1331 [M + H]^+^ (calcd for C_20_H_21_O_6_, 357.1333). Its IR absorptions at 3444, 1633, and 1509 cm^−1^ indicated the presence of hydroxy, olefinic, and aromatic functionalities, respectively. In the ^1^H NMR of 1, a 1,3,4,5-tetrasubstituted aromatic moiety [δ_H_ 7.07 (d, *J* = 1.8 Hz, H-6) and 7.22 (d, *J* = 1.8 Hz, H-2)], an ABX-type aromatic functionality [δ_H_ 6.69 (dd, *J* = 7.9, 1.8 Hz, H-6′), 6.74 (d, *J* = 7.9 Hz, H-5′) and 6.87 (d, *J* = 1.8 Hz, H-2′)], two trans-mutual coupled olefinic protons [δ_H_ 6.33 (d, *J* = 15.9 Hz, H-8) and 7.56 (d, *J* = 15.9 Hz, H-7)], a terminal allylic group [δ_H_ 4.99 (ddd, *J* = 17.1, 1.8, 1.8 Hz, H-9′a), 5.10, (br d, *J* = 7.6 Hz, H-7′), 5.15 (ddd, *J* = 10.1, 1.8, 1.8 Hz, H-9′b) and 6.40 (ddd, *J* = 17.1, 10.1, 7.6 Hz, H-8′)] and two methoxyl resonances [δ_H_ 3.77 (s, 3′-OCH_3_) and 3.92 (s, 3-OCH_3_)] were observed. Twenty carbon resonances, attributable to seven non-protonated aromatic carbons [δ_C_ 126.7 (C-1), 131.2 (C-5), 135.3 (C-1′), 146.1 (C-4′), 148.6 (C-3), 147.2 (C-4) and 148.2 (C-3′)], one acid carbonyl (δ_C_ 168.4, C-9), one methine (δ_C_ 48.2, C-7′), eight olefinic methines [δ_C_ 108.9 (C-2), 113.2 (C-2′), 115.6 (C-5′), 116.1 (C-8), 121.8 (C-6′), 123.8 (C-6), 141.5 (C-8′) and 146.2 (C-7)], one exomethylene (δ_C_ 115.9, C-9′) and two methoxyls [δ_C_ 56.4 (3′-OCH_3_) and 56.6 (3-OCH_3_)], were observed in the ^13^C NMR spectrum coupled with the DEPT spectrum of **1** (Table 1). The connectivity of **1** was further deduced by cross-peaks of δ_H_ 5.10 (H-7′)/δ_C_ 113.2 (H-2′), 115.9 (H-9′), 121.8 (H-6′), 123.8 (C-6), 131.2 (C-5), 135.3 (C-1′), 141.5 (C-8′) and 147.2 (C-4), δ_H_ 7.56 (H-7)/δ_C_ 108.9 (C-2), 116.1 (C-8), 123.8 (C-6), 126.7 (C-1) and 168.4 (C-9), δ_H_ 3.77 (3′-OCH_3_)/δ_C_ 148.2 (C-3′) and δ_H_ 3.92 (3-OCH_3_)/δ_C_ 148.6 (C-3) in the HMBC spectrum (Figure 2B), which were further corroborated by the mutually-correlated signals of δ_H_ 3.77 (3′-OCH_3_)/δ_H_ 6.87 (H-2′) and δ_H_ 3.92 (3-OCH_3_)/δ_H_ 7.22 (H-2) in the NOESY spectrum (Figure 2B). Accordingly, **1** was characterized as shown, and was named as 5-[3-(4-hydroxy-3-methoxyphenyl)allyl]ferulic acid. To our knowledge, **1** with two sets of C_6_–C_3_ unit connected at C-7′ was a new skeletal type of lignan.

Compound **2** was isolated as colorless oil with molecular formula C_11_H_20_O_2_ as deduced by positive-ion HR-ESIMS, showing an [M + H]^+^ ion at *m/z* 185.1501 (calcd for C_11_H_21_O_2_, 185.1541). Conspicuous absorptions at 3445 and 1660 cm^−1^ in the IR spectrum of **2** indicated the presence of hydroxy and olefinic functionalities, respectively. The ^1^H NMR (Table 2) coupled with COSY spectrum of **2** showed two aliphatic chains at δ_H_ 0.85–1.97 (–H_2_-7–H_3_-11) and δ_H_ 1.66–5.44 (–H-3–H-2–H-1–H_2_-6–H_2_-5–). The above assignments also reflected in the ^13^C NMR of **2** supported by DEPT spectra, in which one methyl (δ_C_ 14.2, C-11), six methylenes [δ_C_ 22.7 (C-10), 26.3 (C-6), 27.2 (C-5), 27.4 (C-8), 31.8 (C-9) and 37.4 (C-7)], three methines [δ_C_ 67.1(C-2), 69.2 (C-1) and 121.0 (C-3)] and one quaternary carbon (δ_C_ 144.1, C-4) were observed. In the HMBC spectrum of 2 (Figure 2C), cross-peaks of δ_H_ 5.44 (H-3)/δ_C_ 27.2 (C-5) and 37.4 (C-7), δ_H_ 1.92–2.01 and 2.04–2.10 (H2-5)/δ_C_ 144.1 (C-4) and δ_H_ 1.97 (H2-7)/δ_C_ 144.1 (C-4) indicated C-1–C-6 was a cyclohexene moiety with a double bond at Δ^3^, and C-7–C-11, a saturated linear aliphatic chain, was attached at C-4. The relative configurations of hydroxy-bearing chiral C-1 and C-2 were approached by the *J* values of carbinoyl protons H-1 (9.1, 4.2 Hz) and H-2 (4.2 Hz), which indicated that H-1 and H-2 were pseudo-axial- and pseudo-equatorial-oriented, respectively (Figure 2C). The relative configuration of 1,2-dihydroxy in **2** was thus deduced to *cis*. Conclusively, the structure of **2** was elucidated as shown, and was named as *cis*-4-pentylcyclohex-3-ene-1,2-diol.

### 2.2. Effects of Compounds ***1*** and ***8*** on Melanin Content in α-MSH-Stimulated B16-F10 Cells

To ascertain the depigmentation constituents in rhizoma of *L. sinense*, all the pure isolates were subjected to antimelanogenesis assay in B16-F10 melanoma cells. Murine melanoma B16-F10 cells are a well-established model for antimelanogenic principles discovery, and have been widely adopted in previous studies [48]. α-Melanocyte-stimulating hormone (α-MSH) is a peptide hormone and responsible for the production of melanin by melanocytes through activating melanocortin **1** receptor [49]. In this research, B16-F10 cells were stimulated with α-MSH (100 nM) and simultaneously treated with each compound at concentrations of 25, 50, or 100 μM. The melanin content of B16-F10 melanoma cells without compound treatment was assigned as 100%. Arbutin, a common skin whitening agent in cosmetic products, was used as positive control. Of these compounds, **1** and **8** inhibit α-MSH-induced melanin production in a dose-dependent manner. The IC_50_ values of compounds **1** and **8** were 78.9 and 31.1 μM, respectively (Figure 3B). Compound **1** and **8** at the effective concentrations did not show obvious effects on MTT assay (Figure 3A). The MTT results may arise from the combined effects of cell proliferation reduction and cell viability inhibition according to the experimental conditions. These results suggested that the anti-melanogenic effects of compound **1** and **8** did not attribute to the cell death or growth inhibition.

Using the HPLC-DAD method, the main effective constituents (**1** and **8**) were characterized from the crude extract (Figure 4) by comparing with the retention time of pure **1** (from laboratory synthesis, data not published yet) and **8** (from Sigma) as standards. Stock solutions of 1, 10, 20, 40, 60, 80 and 100 µg/mL were utilized. Each concentration was injected in triplicate. The content ratios of **1** and **8** in the extract of dried material were quantified to be 0.009% and 0.15% (*w*/*w*) by linear regression of the respective peak areas.

### 2.3. In Vivo Zebrafish Pigmentation Assay

In addition to evaluate the effect of 8 on in vitro antimelanogenesis, its in vivo anti-pigmentation ability through zebrafish pigmentation assay was further investigated. Zebrafish has been considered as an advantageous vertebrate model organism due to its small size, high fecundity, and similar gene sequences and organ systems to human beings. Additionally, the melanin pigmentation process on its surface allowing easy observation makes zebrafish a particular useful model for investigating in vivo melanogenic inhibitors or stimulators [50]. In this study, arbutin at 20 mM was used as positive control, and arbutin of 20 μM was included to compare with compound **8** at same concentration level. After the incubation of zebrafish embryos from 7 hpf to 72 hpf, compound **8** exhibited higher pigmentation inhibitory activity at concentrations of 10 and 20 μM compared to arbutin (20 μM) (Figure 5). Upon the treatment of 20 μM compound **8**, the pigmentation level of zebrafish markedly decreases about 31%; while compound **8** at 10 μM decreases 26.2% pigmentation level.

### 2.4. Viability Assay of Human Epidermal Skin Equivalents

The safety of cosmetic products is a serious concern. For instance, hydroquinone, an effective skin lightening agent, has been banned from the market because of numerous adverse reactions and controversy over the potential carcinogenic risk [7]. Kojic acid, a potent tyrosinase inhibitor, may induce contact dermatitis and potential genotoxicity [51,52]. In order to assess the safety of compounds **1** and **8** on human skin, we performed a cell viability assay on Human skin equivalents (HSEs) model. HSEs are three-dimensional culture systems that are generated by seeding human keratinocytes onto an appropriate dermal substrate pre-seeded with human fibroblasts. HSEs are physiologically comparable to the natural skin and provide suitable alternatives for animal testing. Under controlled culture conditions, the HSEs demonstrate high similarity with the native tissue from which it was derived [53]. In this study, viability assays on the normal human epidermal keratinocytes (NHEKs) in Leiden epidermal models (LEMs) were performed. Compounds **1** and **8** did not affect the cell viability at the concentrations of 10–100 μM. The cell viabilities of compound **1** are 98% and 106 % at the concentrations of 10 and 100 μM, respectively. The cell viabilities of compound **8** are 97% and 92% at the concentrations of 10 and 100 μM, respectively. Accordingly, compounds **1** and **8** did not exert cytotoxicity against NHEKs in the Leiden epidermal models at concentrations below 100 μM (Figure 6). Since the effective concentration of **8** is below 100 μM, which suggests that **8** might be safe for skin whitening below the tested concentrations. However, in practice, cosmetic products may be applied on human skin for a long time, thus an extended experimental period of tested compounds on HSEs has to be further conducted.

### 2.5. Molecular Docking Study of B16-Mus Musculus Tyrosinase

Tyrosinase catalysis is the rate-limiting step of melanin biosynthesis. Thus, inhibition of tyrosinase is the most common approach to achieve skin whiteness [54,55]. In order to determine whether *L. sinense* suppressed melanogenesis in B16-F10 cells through tyrosinase inhibition, 3D stick models (Figure 7A) and a 2D diagram (Figure 7B) of molecular docking using DS software were performed, which revealed the possible inhibitory mechanism of compound **8** on mouse (Mus musculus) tyrosinase. It was shown that the active site of mouse tyrosinase was located at a domain surrounded by amino acids His377, Asn378, His381, Gly389, Thr391, Ser394 together with two copper ions. The CDOCKER interaction energy between the enzyme and inhibitor was −46.0067 kcal/mol. The simulation results showed that hydrophobic amino acids including Gly389, Asn 378, Thr391 Ser394, His 404, and His192 around the catalytic site formed van der Waals forces with compound **8**. Additionally, the oxygen atom of γ-lactone moiety of **8** forms hydrogen bonds with His377 and His215, while its carbonyl group forms coordination bonds with two copper ions. It was also observed that the cyclohexene ring and butane exerted weak π-alkyl interaction with His381 and His215, respectively. Previous studies have demonstrated that *L. sinense* extract displayed mushroom tyrosinase inhibition and down-regulation of tyrosinase mRNA expression in B16-F10 cells [21,56]. Since the molecular docking illustrated a strong interaction between the active domain of mouse tyrosinase and compound **8**, it was thus proposed that (3*S*,3a*R*)-neocnidilide (**8**) exhibited antimelanogenesis activity due to tyrosinase activity attenuation and further decreasing melanin production. However, in addition to tyrosinase inhibition, the other underlying mechanisms of antimelanogenesis activity of (3*S*,3a*R*)-neocnidilide (**8**) remains to be further investigated.

## 3. Materials and Methods 

### 3.1. General

HPLC-grade solvents, *n*-hexane, ethyl acetate, methanol and acetonitrile, were purchased from J. T. Baker (Phillipsburg, NJ, USA). Ethanol was purchased from Merck (Darmstadt, Germany). α-Melanocyte-stimulating hormone (α-MSH), dimethyl sulfoxide (DMSO), phosphate-buffered saline (PBS), 3-(4,5-dimethylthiazol-2-yl)-2,5-diphenyl tetrazolium bromide (MTT) were purchased were purchased from Sigma Aldrich (St. Louis, MO, USA). Open column chromatography was performed on silica gel (70–230 mesh, Merck, Darmstadt, Germany). Pre-coated silica gel plates 60 F254 and Aluminum Sheets RP-18 F254S for TLC were purchased from Merck (Darmstadt, Germany). Optical rotations were measured on a JASCO P-1020 polarimeter (Jasco, Tokyo, Japan). ^1^H and ^13^C NMR were acquired with a Bruker Avance DRX-500 spectrometer (Bruker, Rheinstetten, Germany). Low resolution and high resolution mass spectra were obtained using an ABI API 4000 Q-TRAP ESI-MS (Applied Biosystem, Foster City, CA, USA) and Q-Exactive Plus HR-ESI-MS (Thermo Fisher Scientific, MA, USA), respectively. IR spectra were recorded on a JASCO FT/IR 4100 spectrometer (Jasco, Tokyo, Japan).

### 3.2. Plant Materials

Dried rhizoma of *L. sinense* Oliv. was purchased from Sheng Chang Pharmaceutical Co., Ltd., Taoyuan, Taiwan.

### 3.3. Isolation and Structural Elucidation

Dried rhizoma (9.9 kg) of *L. sinense* was smashed and extracted with methanol (40 L × three times), which was filtered and evaporated to give a black residue (1758 g). This residue was then suspended in H_2_O (3.0 L) and partitioned with equal volume of ethyl acetate and *n*-BuOH for three times, successively. Each layer was concentrated under reduced pressure to obtain EtOAc (415 g), *n*-BuOH (147 g), and H_2_O (896 g) layers. Subsequently, the dried ethyl acetate layer (250 g) was mixed with 375 g silica gel, and was loaded onto a conditioned open column packed with 3550 g silica gel and eluted in a step-wise gradient method by mixtures of *n*-hexane, ethyl acetate and methanol. Each 500 mL was collected for one fraction and analyzed by TLC. Then, all the fractions were combined into eight portions I–VIII according to the results of TLC analyses, which were re-dissolved in a minimum volume of *n*-hexane—ethyl acetate mixtures used in the subsequent HPLC system. Portion II eluted by *n*-hexane—ethyl acetate (95:5) was purified by a semi-preparative HPLC (Hibar® Fertigäute, 10 × 250 mm) using *n*-hexane—ethyl acetate (96:4) as eluent at a flow rate of 3 mL/min to afford **6** (4.2 mg, t_R_ = 19.8 min), **3** (6.1 mg, t_R_ = 21.2 min), **5** (32 mg, t_R_ = 23.5 min) and **7** (79 mg, t_R_ = 28.0 min). The same portion was purified by a semi-preparative HPLC (Phenomenex® Luna, 10 × 250 mm) using *n*-hexane—ethyl acetate (99:1) as eluent at a flow rate of 3 mL/min to afford **4** (36 mg, t_R_ = 32.5 min). Portion III eluted by *n*-hexane—ethyl acetate (90:10) was purified by a semi-preparative HPLC (Phenomenex® Luna, 10 × 250 mm) using *n*-hexane—acetone (95:5) as eluent at a flow rate of 3 mL/min to afford **8** (1.7 g, t_R_ = 19.3 min). Portion IV eluted by *n*-hexane—ethyl acetate (80:20) was purified by a semi-preparative HPLC (Phenomenex® Luna, 10 × 250 mm) using *n*-hexane—ethyl acetate (85:15) as eluent at a flow rate of 3 mL/min to afford **15** (19 mg, t_R_ = 24.4 min), **12** (11 mg, t_R_ = 26.9 min), **9** (25 mg, t_R_ = 33.8 min), **10** (31 mg, t_R_ = 37.0 min), **11** (24 mg, t_R_ = 42.5 min). The same portion was purified by the same column using *n*-hexane—ethyl acetate (78:22) as eluent at a flow rate of 3 mL/min to obtain **14** (10 mg, t_R_ = 20.1 min) and **13** (22 mg, t_R_ = 24.6 min). The same portion was purified by the same column using *n*-hexane—ethyl acetate—acetone (80:10:10) as eluent at a flow rate of 3 mL/min to obtain **20** (25 mg, t_R_ = 13.2 min), **17** (15 mg, t_R_ = 16.3 min), **18** (28 mg, t_R_ = 18.5 min), **16** (132 mg, t_R_ = 21.8 min) and **19** (39 mg, t_R_ = 25.6 min). Portion V eluted by *n*-hexane—ethyl acetate (60:40) was purified by a semi-preparative HPLC (Hibar® Fertigäute, 10 × 250 mm) using *n*-hexane—ethyl acetate—acetone (68:27:5) as eluent at a flow rate of 3 mL/min to afford **21** (5.3 g, t_R_ = 10.2 min), **23** (63 mg, t_R_ = 20.0 min), **22** (84 mg, t_R_ = 22.0 min) and **24** (33 mg, t_R_ = 26.5 min). The same portion was purified by a semi-preparative HPLC (Hibar® Fertigäute, 10 × 250 mm) using *n*-hexane—ethyl acetate (72:28) as eluent at a flow rate of 3 mL/min to afford **2** (24 mg, t_R_ = 24.3 min). Portion VI eluted by *n*-hexane—ethyl acetate (40:60) was purified by a semi-preparative HPLC (Hibar® Fertigäute, 10 × 250 mm) using *n*-hexane—ethyl acetate (53:47) as eluent at a flow rate of 3 mL/min to afford **1** (47 mg, t_R_ = 13.2 min).

### 3.4. Spectroscopic Data

5-[3-(4-Hydroxy-3-methoxyphenyl)allyl]ferulic acid (**1**): colorless oil; [α]${}_{D}^{25}$ +5.6° (*c* 0.12, CH_3_OH); IR (neat) *ν*_max_ 3444, 2935, 1633, 1509, 1434, 1376, 1267, 1153; positive ESI-MS *m/z* 357.2 [M + H]^+^; positive HRESI-MS *m/z* 357.1331 [M + H]^+^ (calcd for C_20_H_21_O_6_, 357.1333); ^1^H and ^13^C NMR data see Table 1.

*Cis*-4-pentylcyclohex-3-ene-1,2-diol (**2**): colorless oil; [α]${}_{D}^{22}$ ‒20.3° (*c* 0.20, CH_3_OH); IR (neat) *ν*_max_ 3445, 2919, 1660, 1455, 1371, 1225, 1084; ESIMS *m/z* 185.2 [M + H]^+^; HRESI-MS *m/z* 185.1501 [M + H]^+^ (calcd for C_11_H_21_O_2_, 185.1541); ^1^H and ^13^C NMR data see Table 2.

### 3.5. HPLC–DAD Analysis

Chromatographic analyses were performed on a Hitachi HPLC system consisting of L-7100 pump, L-7200 autosampler, L-7455 detector and D-7000 system manager data acquisition software, on an XBridge^TM^ C18 column (250 mm length, 4.6 mm internal diameter, 5 um particle size; Waters). The mobile phase consisted of H_2_O (solvent A) and CH_3_CN (solvent B). The flow rate was 1.0 mL/min. The elution program was as follows: isocratic with 2% B (0–5 min), 2–20% B (5–25 min), 20–90% B (25–30 min), and isocratic with 90% B (30–35 min). The injection volume was 10 L. UV–visible spectra were recorded at 240 nm.

### 3.6. Cell Culture

The B16-F10 murine melanoma cells (CRL6475) were purchased from the Food Industry Research and Development Institute (FIRDI, Hsinchu, Taiwan). The cells were cultured in 90% Dulbecco’s Modified Eagle’s Medium (DMEM, Sigma-Aldrich, St. Louis, MO, USA) supplemented with 10% fetal bovine serum (FBS, Sigma, St. Louis, MO, USA) and 1% penicillin–streptomycin solution in culture flasks in a CO_2_ incubator with a humidified atmosphere containing 5% CO_2_ in the air at 37 °C. The culture medium was changed every two days. The cells were harvested by trypsinization when they were about 85% confluent, counted with a haemocytometer (Neubauer Improved., Marienfeld, Germany) and seeded at the appropriate numbers into wells of cell culture plates for further experiments.

### 3.7. Cell Viability Assay

To determine the safety of the various extracts the viability of cells following treatment with extracts was determined by the MTT assay. This method is based on the reduction of 3-(4,5-dimethylthiazol-2-yl)-2,5-diphenyl tetrazolium bromide (MTT) to formazan by mitochondrial enzymes in viable cells [57]. The quantity of formazan formed is proportional to the number of viable cells present and can be measured spectrophotometrically. Briefly, 100 nM α-melanocyte stimulating hormone (α-MSH)-pretreated cells seeded at a density of 1 × 10^4^ cells/well in a 12-well plate were left to adhere overnight. Pure isolates or arbutin were then added to each well and incubated for another 72 h. Then, the treated cells were labelled with MTT dye reagent (Applichem, Denmark) in PBS (2 mg/mL) for 3 h. The formazan precipitates were dissolved by DMSO and the concentrations were measured at 570 nm in a microplate reader. Cell viability was calculated using the following formula: cell viability (%) = (A sample/A control) × 100, where A sample and A control are the absorbances from the mixture with, or without the addition of test sample, respectively.

### 3.8. Melanin Content Assay

Melanin content was measured as described previously with slight modifications [58]. The B16-F10 melanoma cells were seeded with 1 × 10^4^ cells/well in 3 mL of medium in 6-well culture plates and incubated overnight to allow cells to adhere. The cells were exposed to various concentrations (25, 50 and 100 μM) of the pure isolates or arbutin for 72 h in the presence of 100 nM α-MSH. At the end of the treatment, the cells were washed with PBS and lyzed with 150 μL of 1 N NaOH (Merck, Germany) containing 10% DMSO for 1 h at 80 °C. The absorbance at 405 nm was measured using a microplate reader. The melanin content of B16-F10 melanoma cells without compound treatment was assigned as 100%, and the melanin content of compound-treated cells was calculated relative to the control group.

### 3.9. In Vivo Zebrafish Pigmentation Assay

The animal use protocol has been reviewed and approved by the Institutional Animal Care and Use Committee or Panel (IACUC/IACUP) of Taipei Medical University (No. LAC-2017-0311). The methods were carried out in compliance with the relevant laws and the approved guidelines. Wild-type zebrafish embryos were collected from Zebrafish Core Facility of Taipei Medical University. Phenotype-based evaluation of zebrafish embryo was performed according to the previous study with slight modification [50]. The embryos were incubated at 28 °C with 1% ethanol as control and a different concentration of compounds from 7 hpf (post-fertilization) to 72 hpf. To evaluate the anti-melanogenesis effects of melanogenic modulators on zebrafish developmental process, the pigmentation of zebrafish was analyzed at 72 hpf. The embryos were mounted in 1% Low Melting Agarose (Bioshop Canada, Burlington, ON, Canada) and captured images with a ZEISS Stemi 508 stereomicroscope (ZEISS, Oberkochen, Germany). Images pixel measurement analysis was carried out by Fiji package of ImageJ (http://rsb.info.nih.gov/ij/index.html, NIH, USA). The quantification of pigmentation data was analyzed (the area below the eyes of the zebrafish) and compared to control group.

### 3.10. Viability Assay on Normal Human Epidermal Keratinocytes

All primary human skin cells from healthy donors used by the Department of Dermatology of the Leiden University Medical Center are isolated from surplus tissue collected according to article 467 of the Dutch Law on Medical Treatment Agreement and the Code for proper Use of Human Tissue of the Dutch Federation of Biomedical Scientific Societies. According to article 467 surplus tissues can be used if no objection is made by the patient. This means that the patient who will undergo plastic surgery is well informed on the research. None of the authors were involved in the tissue sampling. The Declaration of Helsinki principles were followed when working with human tissues.

The fresh mamma reduction surplus skin of a single female individual was used for isolation of normal human epidermal keratinocytes (NHEKs) as described previously [59]. The NHEKs in Leiden epidermal models (LEMs) were incubated overnight under submerged conditions in keratinocyte medium. Within four days, fetal bovine serum was gradually omitted and the NHEKs in LEMs were cultured serum-free at the air-liquid interface for seven days, while culture medium was refreshed twice a week. Viability assays were performed by adding 0.5 mL of 1 mg/mL MTT to each of the NHEKs in LEMs for 3 h, after 24 h exposure to the test compounds **1**, **8**, and DMSO (negative control). The precipitated blue formazan product was extracted from the cells within 2 h with 0.5 mL isopropanol per well. The concentration of formazan was measured by determining the OD at 570 nm using a Tecan Infinite F50 microplate reader.

### 3.11. Statistical Analysis

All the data in our study were obtained as averages of experiments that were performed at least in triplicate and expressed as means ± SD (Standard deviation). Statistical analysis was performed by Student’s t-test. The statistical significance of results was set at *p* < 0.05 (*) and *p* < 0.01 (**).

### 3.12. Molecular Docking Study of B16-Mus Musculus Tyrosinase

#### 3.12.1. Homology Modeling

As no three-dimensional structures for Mus musculus tyrosinase are available now, Homology modeling is the most assured method for prediction of three-dimensional structures of unknown protein based on the assumption that the structure of the unknown protein is similar to the known structures of some homologous reference proteins. We acquired the Mus musculus tyrosinase amino acid sequence from the National Center for Biotechnology Information (NCBI, https://www.ncbi.nlm.nih.gov/) protein sequence database. A homolog protein template of the query protein Mus musculus tyrosinase was identified by Phyre2 (Protein Homology/analog Y Recognition Engine V 2.0) [60]. The 446 residues (81% of Mus musculus sequence) have been modelled with 100.0% confidence by the single highest scoring template as PDB code: c5m8pA was chosen as a receptor for the docking calculation studies.

#### 3.12.2. Analysis of Ligand-Protein Interaction 

The binding site of the Mus musculus tyrosinase was determined based on reference human tyrosinase (PDB 5M8M) six amino acid residues and the binding pocket have two copper. Therefore, the binding site sphere was defined. Subsequently, the 3D structure of the compound **8** was prepared and optimized by energy minimization docked into the binding pocket of the Mus musculus tyrosinase using the CDOCKER program and the number of docking runs was set to 50 for inhibitor. All other parameters were set as default to the analysis of Ligand-Protein Interaction through using Biovia Discovery Studio v.4.0 (Accelrys Software Inc., San Diego, CA, USA). Finally, from the 50 docking conformations, the top one with the highest CDOCKER energy score was chosen to explore the binding mode of docked compound in the Mus musculus tyrosinase active site.

## 4. Conclusions

In the present study, we adopted a bioassay-guided method using B16-F10 cells to isolate the antimelanogenesis constituents from *L. sinense* rhizoma extracts. The active constituents were determined to be 5-[3-(4-hydroxy-3-methoxyphenyl)allyl]ferulic acid (**1**) and (3*S*,3a*R*)-neocnidilide (**8**). According to the HPLC analysis, the content of compound **8** accounts for 0.15% of the crude extract. The antimelanogenesis activity of **8** was verified by both in vitro B16-F10 cells and in vivo zebrafish assays. All these findings suggested that **8** may, at least, provide a rationale for the potential antimelanogenesis effect of *L. sinense* for its high potency and quantity. The cell viability data on B16-F10 cells and NHEKs imply that **8** could be developed potentially as an antimelanogenesis agent. The mode of action of **8** on antimelanogenesis was speculated to be the inhibition of the tyrosinase activity based on the results of molecular docking; however, that remains to be further confirmed. Our finding revealed that compound **8** exhibited anti-melanogenesis effects and safety through in vitro, in vivo, and ex vivo studies, however, the depigmenting efficacy and biosafety on human skin need to be evaluated and validated by clinical trials in the future.

## 5. Patents

An earlier version of the manuscript has been published as a patent (patent number: EP 2832719, TW I507390).

## Figures and Tables

**Figure 1 ijms-19-03994-f001:**
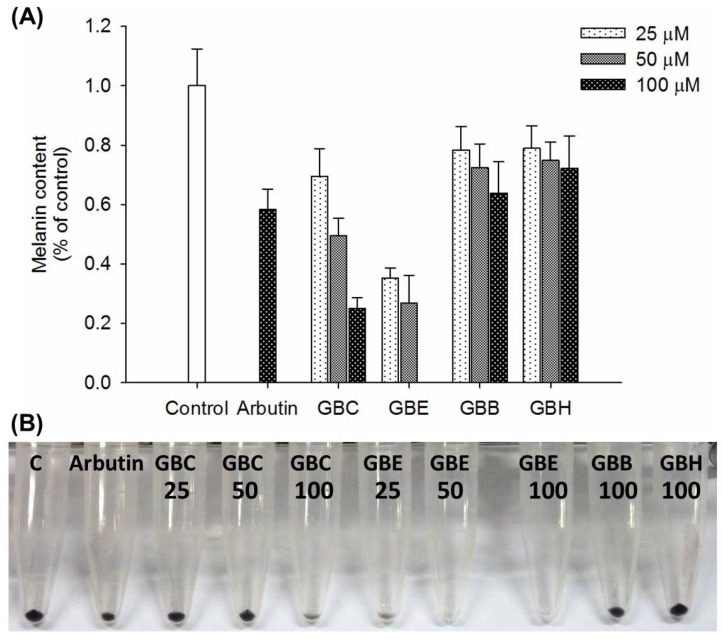
(**A**) The antimelanogenesis effect of different extraction layers with particular concentration on B16-F10 melanoma cells. The B16-F10 melanoma cells were seeded and incubated overnight to allow cells to adhere. The cells were exposed to various concentrations (25, 50 and 100 μM) of the different extraction layers or arbutin for 72 h in the presence of 100 nM α-MSH. At the end of the treatment, the cells were washed with PBS and lyzed with 150 μL of 1 N NaOH containing 10% DMSO for 1 h at 80 °C. The absorbance at 405 nm was measured using a microplate reader. (**B**) Melanin contents in lyzed B16-F10 melanoma cells of vehicle control (C), positive control (arbutin 100 μM), treatments of crude extract (GBC, 25, 50 and 100 μg/mL), ethyl acetate layer (GBE, 25, 50 and 100 μg/mL), *n*-BuOH layer (GBB, 100 μg/mL), and H_2_O layer (GBH, 100 μg/mL) from rhizoma of *L. sinense*.

**Figure 2 ijms-19-03994-f002:**
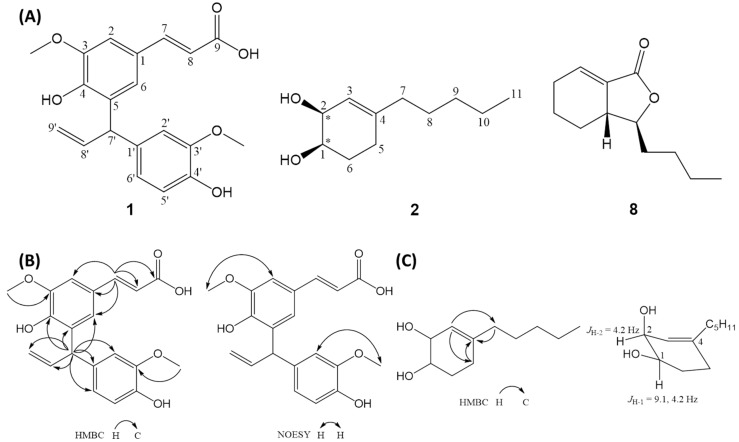
(**A**) Chemical structures of compounds **1**, **2**, and **8** isolated from the rhizoma of *L. sinense*. (**B**) Selected HMBC and NOESY of **1**. (**C**) Selected HMBC and half-chair conformation of **2**.

**Figure 3 ijms-19-03994-f003:**
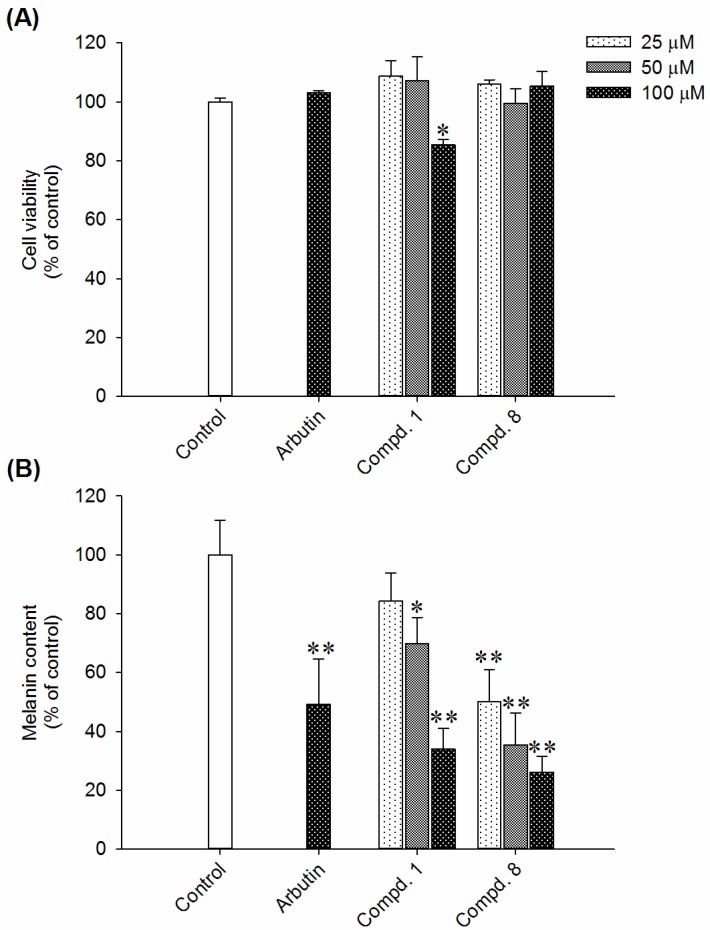
(**A**) The effects of compounds **1** and **8** on cell viability determined by MTT assay. Melanoma cells pretreated with 100 nM α-MSH were seeded at a density of 1 × 10^4^ cells/well in a 12-well plate. Then, the melanoma cells were left to adhere overnight. Pure isolates (25, 50 and 100 μM) or arbutin (100 μM) were added to each well and incubated for another 72 h. Subsequently, the treated cells were labelled with MTT dye reagent in PBS (2 mg/mL) for 3 h. The formazan precipitates were dissolved by DMSO, and the concentrations were measured at 570 nm in a microplate reader. (**B**) The effects of compounds **1** and **8** on melanin contents in B16-F10 cells. Melanoma cells were seeded at a density of 1 × 10^4^ cells/well in a 6-well plate and incubated overnight. The cells were exposed to various concentrations (25, 50 and 100 μM) of the pure isolates or arbutin for 72 h in the presence of 100 nM α-MSH. The cells were washed with PBS and lyzed with 150 μL of 1 N NaOH containing 10% DMSO for 1 h at 80 °C. The absorbance at 405 nm was measured using a microplate reader. Results were expressed as % control and data mean ± S.D. *n* = 3 in each group. * *p* < 0.05, ** *p* < 0.01 compared to the control group.

**Figure 4 ijms-19-03994-f004:**
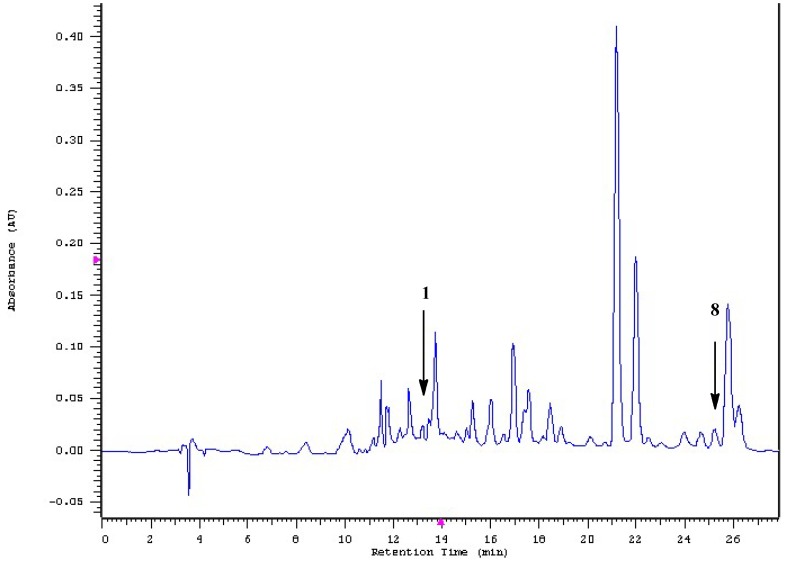
The HPLC-DAD chromatogram of compounds **1** and **8** in the crude extract of rhizoma of *L. sinense*.

**Figure 5 ijms-19-03994-f005:**
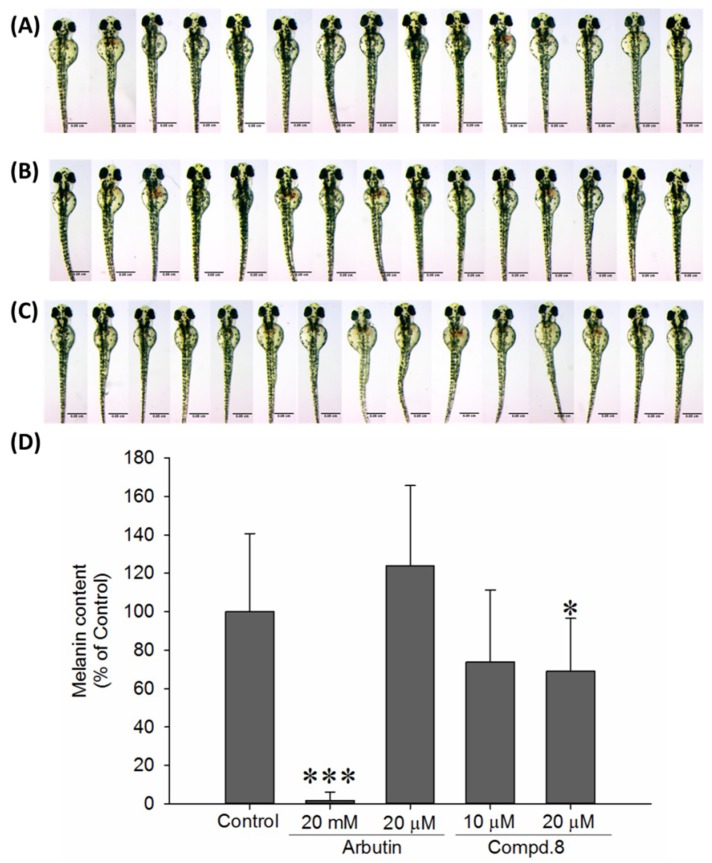
Depigmenting effect of compound **8** and melanogenic regulators on melanogenesis of zebrafish in an in vivo phenotype-based system. Zebrafish embryos were exposed to (**A**) E3 buffer (containing 1% alcohol), (**B**) arbutin (20 μM), or (**C**) compound **8** (20 μM) from 7 hpf (post-fertilization) to 72 hpf. (**D**) Pigmentation levels of zebrafish treated with arbutin (20 mM and 20 μM) and compound **8** (10 μM and 20 μM). Results were expressed as % control and data mean ± S.D. *n* = 15 in each group. * *p* < 0.05 and *** *p* < 0.001 compared to the control group.

**Figure 6 ijms-19-03994-f006:**
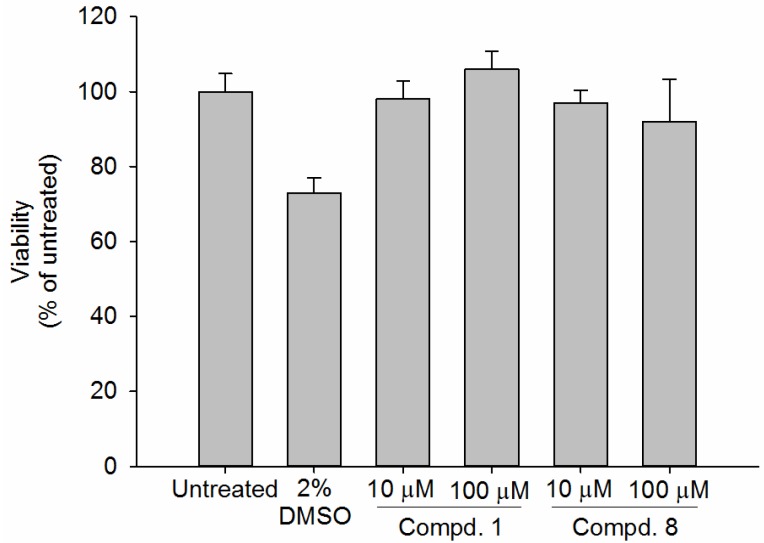
The effect of compounds **1** and **8** on cell viability in normal human epidermal keratinocytes (NHEKs). NHEKs were exposed to **1**, **8**, and vehicle at the indicated concentrations for 24 h. Cell viability was determined by MTT assay.

**Figure 7 ijms-19-03994-f007:**
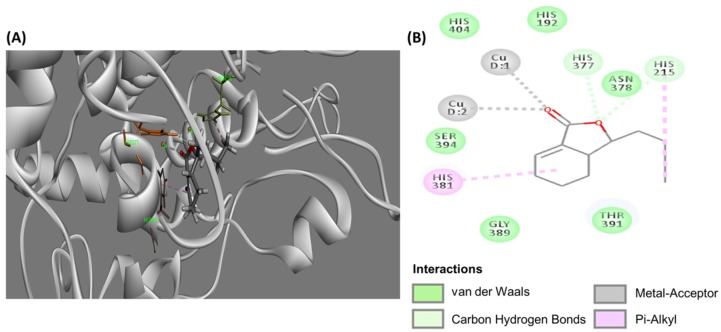
Molecular docking of compound 8 to Mus musculus tyrosinase. Binding conformations of (**A**) 3D stick model and (**B**) 2D diagram of molecular interactions to the active site.

**Table 1 ijms-19-03994-t001:** ^13^C (125 MHz), ^1^H NMR (500 MHz), and HMBC data for compound **1** (in acetone-*d*_6_, δ in ppm).

Position	^13^C NMR *^a^*	^1^H NMR	HMBC
	δ_C_ (multi.)	δ_H_ (multi., *J* in Hz)	H→C
1	126.7 (s)		
2	108.9 (d)	7.22 (d, 1.8)	C-1, C-3, C-4, C-6
3	148.6 (s)		
4	147.2 (s)		
5	131.2 (s)		
6	123.8 (d)	7.07 (d, 1.8)	C-2, C-4, C-7′
7	146.2 (d)	7.56 (d, 15.9)	C-1, C-2, C-6, C-8, C-9
8	116.1 (d)	6.33 (d, 15.9)	C-1, C-7, C-9
9	168.4 (s)		
1′	135.3 (s)		
2′	113.2 (d)	6.87 (d, 1.8)	C-1′, C-3′, C-4′, C-6′, C-7′
3′	148.2 (s)		
4′	146.1 (s)		
5′	115.6 (d)	6.74 (d, 7.9)	C-1′, C-3′, C-4′
6′	121.8 (d)	6.69 (dd, 7.9, 1.8)	C-2′, C-4′, C-7′
7′	48.2 (d)	5.10 (br d, 7.6)	C-4, C-5, C-6, C-1′, C-2′, C-6′, C-8′, C-9′
8′	141.5 (d)	6.40 (ddd, 17.1, 10.1, 7.6)	C-5, C-1′, C-7′
9′	115.9 (t)	4.99 (ddd, 17.1, 1.8, 1.8)	C-7′, C-8′
		5.15 (ddd, 10.1, 1.8, 1.8)	C-7′
3-OCH_3_	56.6 (q)	3.92 (3H, s)	C-3
3′-OCH_3_	56.4 (q)	3.77 (3H, s)	C-3′

*^a^* Multiplicities were obtained from DEPT experiments.

**Table 2 ijms-19-03994-t002:** ^13^C (125 MHz), ^1^H NMR (500 MHz) and HMBC data for compound **2** (in chloroform-*d*, δ in ppm).

Position	^13^C NMR *^a^*	^1^H NMR	HMBC
	δ_C_ (multi.)	δ_H_ (multi., *J* in Hz)	H→C
1	69.2 (d)	3.73 (dt, 9.1, 4.2)	C-3, C-5
2	67.1 (d)	4.08 (br t, 4.2)	C-1, C-3, C-4, C-6
3	121.0 (d)	5.44 (m)	C-1, C-2, C-5, C-7
4	144.1 (s)		
5	27.2 (t)	1.92–2.01 (m)	C-1
		2.04–2.10 (m)	C-1, C-3, C-4, C-6, C-7
6	26.3 (t)	1.66–1.78 (m)	C-1, C-2, C-5
7	37.4 (t)	1.97 (br t, 7.3)	C-4, C-9
8	27.4 (t)	1.38 (m)	C-4, C-7, C-9, C-10
9	31.8 (t)	1.20 (m)	C-7, C-10
10	22.7 (t)	1.27 (m)	C-8, C-9, C-11
11	14.2 (q)	0.85 (t, 7.0)	C-9, C-10

*^a^* Multiplicities were obtained from DEPT experiments.

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
