# Peer review of "Melanogenesis Inhibitors from the Rhizoma of Ligusticum Sinense in B16-F10 Melanoma Cells In Vitro and Zebrafish In Vivo"

_ijms, 2018, doi:10.3390/ijms19123994_

Reviewer 1 Report

In this study, the authors set out to identify compounds within the rhizoma of Ligusticum sinense, a chinese medical plan that are responsible for anti-melanogenesis. The authors identified two compounds 1 and 8 that seem to exhibit anti-melanogenesis activity with no effect on viability. Through 3D modeling the authors hypothesize that compound 8 works by inhibiting the activity of tyrosinase in the B16F10 cells used in this study leading to its anti-melanogenesis activity.  Overall, this study is good; however there are a few areas that should be addressed prior to publication -

Figure 3 - the authors use the MTT assay as a measure of cell viability, which is acceptable; however this assay does not indicate if there is an affect on proliferation of the cells.  If the arbutin, compound 1 or compound 8 also decreased proliferation rates, this could lead to a decrease in melanin content that is not dependent on melanogensis but instead, cell density. It is unclear if cells were counted prior to harvesting for melanin quantification or if these compounds affect proliferation of the cells.

Figure 5 - It is not clear from the images of the zebrafish if there is a decrease in pigmentation after compound 8 treatment.  Perhaps it is the quality of the images, but one would expect to be able to see the decrease in pigmentation as quantified in graph below. The authors are reporting up to a 30% decrease in pigmentation, but this is unclear in the images. Also, it appears as though arbutin actually increases melanogenesis in this system.  The authors were using arbutin as a positive control in their other experiments, but in the model system it does not appear to be a suitable positive control.  This data set needs to be addressed.

Figure 6 - In this figure the authors test the effect of compound 8 on normal skin keratinocytes.  They test for viability after only 24 hours of exposure.  In practice, if compound 8 were to be used in a skin whitening cream it would be applied more frequently then just 24 hours. It is recommended that this study be extended - perhaps to one application per day for one week to determine if there is any real effect on keratinocytes.

In all cases it would be interesting to know if the effects of compounds 1 and 8 are reversible and if so how long does it take to see melanogenesis restored once the compound is removed.

Author Response

Responses to reviewers’ comments

We would like to thank the reviewers for their positive and insightful comments on the manuscript.  We believe that the comments would help to improve the quality of the manuscript, and the responses were raised as below item by item.  All the changed parts were colored red in the manuscript

1. Figure 3 - the authors use the MTT assay as a measure of cell viability, which is acceptable; however this assay does not indicate if there is an effect on proliferation of the cells.  If the arbutin, compound 1 or compound 8 also decreased proliferation rates, this could lead to a decrease in melanin content that is not dependent on melanogensis but instead, cell density. It is unclear if cells were counted prior to harvesting for melanin quantification or if these compounds affect proliferation of the cells.

Response: We agree with the reviewer’s comment.  Based on our experimental designation, the MTT assay was conducted for 72 hours and supplemented with 10% FBS during the whole assay. Such conditions would contribute to cell proliferation more than cell viability, thus the results obtained from such MTT assay could be a mixed effect arose from cell proliferation and cell viability.  We have added the description in the revised manuscript (page 6, lines 13-18)

2. Figure 5 - It is not clear from the images of the zebrafish if there is a decrease in pigmentation after compound 8 treatment.  Perhaps it is the quality of the images, but one would expect to be able to see the decrease in pigmentation as quantified in graph below. The authors are reporting up to a 30% decrease in pigmentation, but this is unclear in the images. Also, it appears as though arbutin actually increases melanogenesis in this system.  The authors were using arbutin as a positive control in their other experiments, but in the model system it does not appear to be a suitable positive control.  This data set needs to be addressed.

Response: In the zebrafish model, the effective concentration of arbutin is usually above 20 mM [1]. In our original version, we did not show this data because we aimed to emphasize the comparison between compound 8 and arbutin at the same concentration level. We reconducted the zebrafish experiments which included the group of arbutin at 20 mM.  We also changed the image quality and increased the sample number from 10 to 15 to exhibit more clearly the comparison between compound 8 and control. The revised figure is displayed on page 9.

3. Figure 6 - In this figure the authors test the effect of compound 8 on normal skin keratinocytes.  They test for viability after only 24 hours of exposure.  In practice, if compound 8 were to be used in a skin whitening cream it would be applied more frequently then just 24 hours. It is recommended that this study be extended - perhaps to one application per day for one week to determine if there is any real effect on keratinocytes.

Response: Actually, we are also interested in the effects of compound 8 on the longer exposure of normal skin keratinocytes; however, the short-term 24-hour toxicity evaluation was just a preliminary study. Based on the result, we may consider conducting a clinical test for its efficiency and safety on human skin in the future study. This information has been added in the “Results and Discussion” section of the revised manuscript (page 9, lines 25–page 10, line 2).

4. In all cases it would be interesting to know if the effects of compounds 1 and 8 are reversible and if so how long does it take to see melanogenesis restored once the compound is removed.

Response: In the present status, we just preliminary evaluate the depigmenting effects and safety of the two compounds. We think the reversibility of the compound effect is slightly far from the scope of this manuscript, although we find some articles studying the reversibility of their compounds on zebrafish model [2]. We will take into consideration to perform this experiment before our future clinical tests.

Reference

[1] Choi, T. Y.; Kim, J. H.; Ko, D. H.; Kim, C. H.; Hwang, J. S.; Ahn, S.; Kim, S. Y.; Kim, C. D.; Lee, J. H.; Yoon, T. J., Zebrafish as a new model for phenotype-based screening of melanogenic regulatory compounds. Pigment Cell Res. 2007, 20, 120-127.

[2] Agalou, A.; Thrapsianiotis, M.; Angelis, A.; Papakyriakou, A.; Skaltsounis, A. L.; Aligiannis, N.; Beis, D., Identification of Novel Melanin Synthesis Inhibitors From Crataegus pycnoloba Using an in Vivo Zebrafish Phenotypic Assay. Front. Pharmacol. 2018, 9, 265.

Reviewer 2 Report

Authors presented the data related to the antimelanogenetic effects of the compounds from rhizoma of Ligusticum sinense.

-Authors used in the study of melanogenesis the murine melanoma B16-F10 cells. Why authors did not
tested the melanogenesis inhibition in human melanocytes? The article would be more valuable and interesting. Authors used human primary keratinocytes from human skin samples, thus such experimental set-up, with primanry melanocytes could be performed. Melanogenesis process in melanoma cells can be disturbed.

-Authors tested the viability on skin equivalents. What about pigmentation testing in those models?

-Please, provide
the description of synthetic melanin solution preparation.
-Authors in the Result section presented data on Human skin equivalents, but in the Material and method
section there is no information about this model.

-Readers would appreciate the more detailed information related to melanin and
it role in physiology and pathology(see and cite Physiol Rev. 2004 Oct;84(4):1155-228).s

-In 2011 the article related to Ligusticum
sinense compounds was published (Zhong Yao Cai. 2011 Mar;34(3):378-80), however authors did not refer to it.

Author Response

Responses to reviewers’ comments

We would like to thank the reviewers for their positive and insightful comments on the manuscript.  We believe that the comments would help to improve the quality of the manuscript, and the responses were raised as below item by item.  All the changed parts were colored red in the manuscript.

1. Authors used in the study of melanogenesis the murine melanoma B16-F10 cells. Why authors did not tested the melanogenesis inhibition in human melanocytes? The article would be more valuable and interesting. Authors used human primary keratinocytes from human skin samples, thus such experimental set-up, with primary melanocytes could be performed. Melanogenesis process in melanoma cells can be disturbed.

Response: Because mouse B16 melanoma cell is a sensitive, reliable, and feasible platform for screening large number of small molecular melanogenesis regulators, we thus applied B16-F10 cell in our studies to test natural compounds from plant extracts.  Furthermore, Zebrafish was found to share similar gene regulation of melanin synthesis with human, and the observation supports the anticipated anti-melanogenic effect on human melanocytes.

The related information was added in the revised manuscript on page 3, line 9-10.

2. Authors tested the viability on skin equivalents. What about pigmentation testing in those models?

Response: We are also interested in the effects of compound 8 on the normal skin keratinocytes; however, the short-term toxicity evaluation was just a preliminary study. Based on the result, we may consider conducting a clinical test for its efficiency and safety on human skin in the future study. This information has been added in the “Conclusion” section of the revised manuscript (page 15, lines 5–7).

3. Please, provide the description of synthetic melanin solution preparation.

Response: After checking the experimental procedure, it was found that we determined the melanin content in comparison with the control group, but not calculated based on a melanin standard curve.  We thus changed the sentence as follows: The melanin content of B16-F10 melanoma cells without compound treatment was assigned as 100%, and the melanin content of compound-treated cells was calculated relative to the control group (page 13, line 22-24).

4. Authors in the Result section presented data on Human skin equivalents, but in the Material and method section there is no information about this model.

Response: In the original version, we described the Human skin equivalents model in section 3.11. Viability Assay on Normal Human Epidermal Keratinocytes. We will move it to section 3.10 in the revised version. Thank you for pointing out the potential for misunderstanding.

5. Readers would appreciate the more detailed information related to melanin and it role in physiology and pathology (see and cite Physiol Rev. 2004 Oct;84(4):1155-228).

Response: Thank you for your valuable comments. We have carefully searched for literatures again and add more information about melanin synthesis and regulation in the “Introduction” section (page 1, line 41 to page 2, line 29). The article “Physiol Rev. 2004 Oct;84(4):1155-228” was cited as reference 16. In 2011 the article related to Ligusticum sinense compounds was published (Zhong Yao Cai. 2011 Mar;34(3):378-80), however authors did not refer to it.

Response: We have cited this reference as reference 16 in our revised manuscript. (Page 2, line 38)

Round  2

Reviewer 1 Report

The authors appear to have made appropriate additions/changes to their study thus strengthening it. There are still some grammatical errors that need to be fixed throughout the manuscript.

Reviewer 2 Report

Authors corrected thair manuscript and improved it significantly.